# High Rate of Mutational Events in SARS-CoV-2 Genomes across Brazilian Geographical Regions, February 2020 to June 2021

**DOI:** 10.3390/v13091806

**Published:** 2021-09-10

**Authors:** Ueric José Borges de Souza, Raíssa Nunes dos Santos, Fabrício Souza Campos, Karine Lima Lourenço, Flavio Guimarães da Fonseca, Fernando Rosado Spilki

**Affiliations:** 1Laboratório de Bioinformática e Biotecnologia, Campus de Gurupi, Universidade Federal do Tocantins, Gurupi 77402-970, Brazil; uericjose@gmail.com (U.J.B.d.S.); engraissanunes@gmail.com (R.N.d.S.); camposvet@gmail.com (F.S.C.); 2Laboratório de Virologia Básica e Aplicada, Departamento de Microbiologia, Instituto de Ciências Biológicas, Universidade Federal de Minas Gerais, Belo Horizonte 31270-901, Brazil; karine_lourenco@hotmail.com (K.L.L.); fdafonseca@icb.ufmg.br (F.G.d.F.); 3Laboratório de Saúde Única, Feevale Techpark, Universidade Feevale, Av. Edgar Hoffmeister, 600, Zona Industrial Norte, Campo Bom 93700-000, Brazil; 4Laboratório de Microbiologia Molecular, Universidade Feevale, Rodovia ERS-239, 2755, Prédio Vermelho, Piso 1, sala 103, Vila Nova, Novo Hamburgo 93525-075, Brazil

**Keywords:** SARS-CoV-2, variants hotspot, genome analysis, viral evolution, mathematical correlation

## Abstract

Brazil was considered one of the emerging epicenters of the coronavirus pandemic in 2021, experiencing over 3000 daily deaths caused by the virus at the peak of the second wave. In total, the country had more than 20.8 million confirmed cases of COVID-19, including over 582,764 fatalities. A set of emerging variants arose in the country, some of them posing new challenges for COVID-19 control. The goal of this study was to describe mutational events across samples from Brazilian SARS-CoV-2 sequences publicly obtainable on Global Initiative on Sharing Avian Influenza Data-EpiCoV (GISAID-EpiCoV) platform and to generate indexes of new mutations by each genome. A total of 16,953 SARS-CoV-2 genomes were obtained, which were not proportionally representative of the five Brazilian geographical regions. A comparative sequence analysis was conducted to identify common mutations located at 42 positions of the genome (38 were in coding regions, whereas two were in 5′ and two in 3′ UTR). Moreover, 11 were synonymous variants, 27 were missense variants, and more than 44.4% were located in the spike gene. Across the total of single nucleotide variations (SNVs) identified, 32 were found in genomes obtained from all five Brazilian regions. While a high genomic diversity has been reported in Europe given the large number of sequenced genomes, Africa has demonstrated high potential for new variants. In South America, Brazil, and Chile, rates have been similar to those found in South Africa and India, providing enough “space” for new mutations to arise. Genomic surveillance is the central key to identifying the emerging variants of SARS-CoV-2 in Brazil and has shown that the country is one of the “hotspots” in the generation of new variants.

## 1. Introduction

In December of 2019, at Wuhan, China, a novel betacoronavirus was first detected. Coronavirus Disease (COVID-19) [1] has developed into a global pandemic, causing waves of epidemics, infecting over 219 million people and 4.55 million deaths globally by 30 August 2021 [2]. The local profile outbreaks were shaped by measures of restrictions, including lockdown, commerce limitations, and travel control. The viral spreading has led scientists to investigate genomic epidemiology, which plays a central role in characterizing and understanding the emergence of viruses [3,4,5]. The SARS-CoV-2 single-stranded RNA is 29.9 kb in size and has positive coding orientation, encoding four major structural proteins on its 3′ end: spike (S), envelope (E), membrane (M), and nucleocapsid (N). These proteins are essential to virus entry into cells and virus particle formation [6,7].

NGS-based SARS-CoV-2 genome characterizations have revolutionized the scale and depth of variant analysis worldwide. Never before has a viral genome been sequenced so globally in such a short time. Even so, numerous reports revealed potential adaptations of the nucleotide (nt), amino acid (aa), and structural heterogeneity of viral proteins, particularly in the S protein [8]. At the moment, the world’s concern is focused on about four functionally well-defined variants—B.1.1.7 (Alpha), B.1.351 (Beta), P.1 (Gamma), and B.1.617.2 (Delta)—which are associated with viral fitness changes [9,10,11,12].

The mutation rate in SARS-CoV-2 is about 10^4^ replacements of base pairs per year, and possible variations may appear in each replication cycle. In the context of investigating evolutionary events, it is possible to compare single-nucleotide polymorphisms (SNPs) in RNA sequences because mutations in coronaviruses occur from RdRp mistakes during viral genome replication [6,13,14].

The spike protein of SARS-CoV-2 contains an N-terminal S1 subunit and a C-terminal membrane proximal to the S2 subunit. The N-terminal domain (NTD), located in the portion S1A, recognizes carbohydrates, such as sialic acid, and it is responsible for the attachment of the virus to the host cell surface. The receptor-binding domain (RBD) in the S1B portion interacts with the human ACE-2 receptor. Between S1 and S2 there is a PRRA sequence motif that functions as a furin cleavage site. The transmembrane domain has a second cleavage site, which also participates in the viral entry into host cells [15]. The mutation D614G in S, for instance, is a frequently identified mutation that has been associated with increased virus transmissibility and infectivity, and it is possibly one of the origins of the widely prevalent B1.1 branch in many countries. Nonetheless, the mutation does not seem to alter the antigenicity of the S protein [16] and was not associated with any changes in disease severity [13].

In this study, we evaluated mutational events across samples from publicly available SARS-CoV-2 sequences available in GISAID since the beginning of the epidemic (from February 2020 to June 2021) in Brazil. Moreover, we propose a mathematical relation between new mutants versus sequenced genomes. This analysis is fundamental to understanding the changes in the viral genome leading to alterations in viral fitness and transmissibility across the population.

## 2. Materials and Methods

### 2.1. Data Retrieval

Whole genome sequences of SARS-CoV-2 obtained from COVID-19 cases in Brazil were downloaded from the Global Initiative on Sharing Avian Influenza Data-EpiCoV (GISAID-EpiCoV) platform (https://www.gisaid.org/, accessed on 30 June 2021) [17]. Only sequences submitted up to 30 June 2021 and complete genomes (above 29,000 bp) were included. The high coverage filter was also applied to ensure acceptable quality. According to GISAID, high coverage means that only entries with less than 1% of undefined bases (NNNs) and no insertions and deletions unless verified by the submitter are tolerated. Sequences with an unidentified division were also excluded from the final dataset. The sequences were downloaded in FASTA format. The annotated reference genome sequence of the SARS-CoV-2 isolate Wuhan-Hu-1 was retrieved from the NCBI database (Accession Number: NC_045512.2).

### 2.2. Data Processing

A total of 16,953 SARS-CoV-2 complete genome sequences obtained in Brazil were included in the study. The data were grouped according to Brazilian regions (Central-West = 939; Northeast = 1847; North = 921; South = 1913; Southeast = 11,333). Genomes were classified by region and were aligned against the SARS-CoV-2 reference genome using Minimap2 [18] aligner. The SAM files from the alignments were sorted, converted to BAM, and indexed using Samtools V1.9, (The Sanger Institute, Hinxton, UK) [19]. The BAM file was subjected to bcftools mpileup and bcftools call (part of the samtools framework) to call variants and generate genomic VCF files. The bcftools filter was then used to filter called variations and to derive the final VCF file. The Variant Effect Predictor (VEP) was used to assess the functional effects of detected variants on SARS-CoV-2 transcripts [20].

### 2.3. Dynamics of SARS-CoV Clades

Genomic surveillance of SARS-CoV-2 across Brazilian regions was performed using the Nextstrain platform (https://nextstrain.org/ncov, accessed on 30 June 2021), an open-source program that generates updated phylogeny with interactive visualization of publicly available SARS-CoV-2 genomes. The pipeline includes subsampling, alignment, maximum-likelihood phylodynamic analysis, temporal dating of ancestral nodes, discrete trait geographic reconstruction, and results visualization in Auspice [21]. Because of the large number of sequenced genomes, we conducted a subsampling of Brazilian genomes from GISAID by using Nextstrain’s bioinformatics toolkit, which includes python3 scripts for preparing GISAID data for processing by Augur [22]. This was conducted by using the subsample-max-sequence option to randomly sample 300 strains from states that had a large number of sequenced genomes (through the use of the “query” option). The Brazilian virus lineages were identified using Pangolin v3.1.5 as implemented on 25 June 2021 [23]. Additionally, metadata of all SARS-CoV-2 genomes submitted to the GISAID database were accessed on 30 June 2021 by using the complete genomes and high-coverage filter, and the genomic clades were inferred according to its nomenclature system at the time of data collection.

### 2.4. Mathematical Model to Estimate the Rate of Genome Mutants and Global Diversity Rate

A total of 1,070,424 SARS-CoV-2 complete and high-coverage genome sequences were obtained and divided by continent: 11,574 genomes from South America; 14,986 from Oceania; 678,977 from Europe; 60,777 from Asia; 296,009 from Europe; and 8101 from Africa. The data were processed following the conditions below and shown in Appendix A): (I) Calculation of an estimate of how many genomes are hypothetically necessary to obtain a new mutant; (II) Comparison of continents by creating an estimate index for the growing rate of variation; and (III) Comparison of countries from the same regions to identify hotspots of the new variants.

For (I), to calculate an estimate of mutants per sequencing, we put at the center of the test the major sequencing region (Europe) comprising 1,000,285 genomes, identifying a maximum of 956 lineages in 49 countries. To obtain how many genomes are necessary to obtain a new mutant, we used a factor correction to equalize the values of lineages, presuming that all the regions are hypothetically sequenced at the same rate.

The factor correction is: ((ΣGE/ΣG) * (ΣLE/ΣL)), considering:

Sum of total genomes used after filtering = ΣG

Sum of total genomes from Europe = ΣGE

Sum of total lineages = ΣL

Sum of total lineages from Europe = ΣLE

Additionally, we applied the same logic for lineage correction and estimated the number of genomes necessary using a correction rate to identify a new variant using as a determinant variable the maximum of lineages, genomes and country.

For (II) and (III), estimating an index growing variants (IGV) and identifying hotspots:

Lineages per country = l

Sum of total lineages from each region: L

The index was estimated using the Shannon index variations ln(l/L) and sum products from both matrices (comparing each country, estimating the variation inside the region).

## 3. Results

### 3.1. Distribution of SARS-CoV-2 Mutations in Brazil

We evaluated the distribution of SARS-CoV-2 mutations across all five Brazilian geographical regions. In total, 42 SNVs were found across the 16,953 SARS-CoV-2 sequenced genomes (Figure 1 and Figure 2). A total of 32 SNVs were found in genomes obtained from all five Brazilian regions. These mutations were found at positions C241T, T733C, C2749T, C3037T, C3828T, A5648C, A6319G, A6613G, C12778T, C13860T, C14408T, G17259T, C21614T, C21621A, C21638T, G21974T, G22132T, A22812C, G23012A, A23063T, A23403G, C23525T, C24642T, G25088T, T26149C, G28167A, C28512G, A28877T, G28878C, G28881A, G28882A, and G28883C (Figure 1 and Figure 2). Nine SNVs were shared across genomes from three regions (Central-West, Northeast, and South) with allele frequencies ranging from 23.2% (A12964G in Central-West) to 43.5% (C12053T in Northeast). The nine mutations were C100T, T10667G, C11824T, A12964G, C12053T, C28253T, G28628T, G28975T, and C29754T (Figure 2). Only one SNV was shared across genomes from Central-West, North, and South (T29834A), with allele frequency of 32.3%, 26.7%, and 58.2% for each region, respectively (Figure 2).

Amongst the 42 found SNVs, 38 were in coding regions, whereas 2 were in the 5′UTR (C100T and C241T), and the other 2 were in the 3′UTR (C29754T and T29834A). The 11 mutations in the coding regions were synonymous or silent, and 27 mutations are predicted to cause amino acid substitutions (missense variants). Among the 27 missense variants, 44.4% were located in the S gene (12-point mutations: C21614T (L18F), C21621A (T20N), C21638T (P26S), G21974T (D138Y), G22132T (R190S), A22812C (K417T), G23012A (E484K), A23063T (N501Y), A23403G (D614G), C23525T (H655Y), C24642T (T1027I), and G25088T (V1176F)). It is also important to point out that all SNVs detected in the S gene were missense mutations. Furthermore, seven missense mutations (25.9%) were located in the N gene (C28512G (P80R), G28628T (A119S), A28877T (S202C), G28878C (S202T), G28881A (R203K), G28883C (G204R), and G28975T (M234I)). The ORF1ab comprises approximately 67.0% of the genome encoding 16 nonstructural proteins and had a total of 6 (22.6%) missense mutations (C3828T (S1188L), A5648C (K1795Q), T10667G (L3468V), C12053T (L3930F), C14408T (P4715L), and G17259T (E5665D)). The ORF3a and ORF8 accounted for one mutation each; T26149C (S253P) and G28167A (E92K), respectively. In addition, nine synonymous mutations were observed in the ORF1ab (T733C, C2749T, C3037T, A6319G, C11824T, C12778T, A12964G, and C13860T), and only one in the ORF8 gene (C28253T) and N gene (G28882A). Additionally, the gene E encoding the envelope protein, the ORF6, ORF7a, and ORF7b were conserved and did not carry any mutation (Figure 3).

By reassigning sequences to the SARS-CoV-2 lineages according to the Pangolin software analysis, we observed the presence of 61 SARS-CoV-2 lineages across Brazilian regions (Figure 4). The majority of Brazilian genome sequences from GISAID used in this study belonged to lineages Gamma or P.1 (*n* = 10,642; 62.8%), Zeta or P.2 (*n* = 1988; 11.7%), B.1.1.28 (*n* = 1346; 7.9%), B.1.1.33 (*n* = 1275; 7.5%), Alpha or B.1.1.7 (*n* = 416; 2.5%), and P.1.2 (*n* = 316; 1.9%), respectively. A total of 38 lineages were found in the Southeast region of Brazil, whereas 21 were found in the South. A total of 29 lineages were found in the Northeast region; and the North and Central-West regions produced 19 and 18 lineages, respectively.

### 3.2. Distribution of SARS-CoV-2 Clades and Lineages/Variants in Brazil

Regarding the distribution between the five Brazilian regions, sequences from the Southeast region comprised a large proportion of Gamma (8123 genomes), Zeta (941 genomes), B.1.1.28 (707 genomes), B.1.1.33 (644 genomes), and Alpha (374 genomes). In the South region the most prevalent lineage was Gamma (827 genomes), followed by Zeta (336 genomes), B.1.1.33 (301 genomes), B.1.1.28 (283 genomes), and P.1.2 (59 genomes). In the Northeast region, Gamma was also the most prevalent lineage (689 genomes), followed by Zeta (417 genomes), B.1.1 (206 genomes), B.1.1.33 (180 genomes), and B.1.1.28 (145 genomes). In the Central-West region the most prevalent lineages were Gamma (585 genomes), Zeta (158 genomes), B.1.1.28 (69 genomes), B.1.1.33 (63 genomes), and Alpha (23 genomes), respectively. The North region apparently has slightly different dynamics with the Gamma being the most prevalent (418 genomes), followed by B.1.1.28 (142 genomes), Zeta (136 genomes), B.1.1.33 (87 genomes), and B.1.195 (51 genomes) (Figure 4).

Additionally, a state-by-state view shows a high distribution of the Gamma variant of concern (VOC) and the Zeta variant of interest (VOI) across almost all Brazilian states. The Gamma and Zeta lineages were represented by genomes from twenty-six states and were not registered only in the Central-West state of Mato-Grosso, which can be related to the extremely low sequencing rate in this state (see Appendix A).

To date, twenty major clades of SARS-CoV-2 were defined by Nextstrain [21] (19A, 19B, 20A, 20B, 20C, 20D, 20E (EU1), 20F, 20G, 20H (Beta, V2), 20I (Alpha, V1), 20J (Gamma, V3), 21A (Delta), 21B (Kappa), 21C (Epsilon), 21D (Eta), 21E (Theta), 21F (Iota), 21G (Lambda), and 21H), based on global frequency and characteristic mutational events observed in the genomes. A total of 5351 Brazilian genomes representing all states were used for phylogenetic analysis (Figure 5a). The genomes were found to be classified under nine clades (19A, 19B, 20B, 20C, 20D, 20I (Alpha, V1)), 20J (Gamma, V3), 21A (Delta), and 21D (Eta), and the phylogenetic examination showed that the majority belonged to clade 20B (*n* = 2724; 50.91%) and 20J (Gamma, V3) (2516; 47.21%) (Figure 5a). As shown in Figure 5b, this was probably a reflection of the sequenced genomes from the beginning of the pandemic.

It is important to point out that different nomenclatures for SARS-CoV-2 have been proposed, including by Nextstrain, cov-lineages.org [23], and GISAID [17]. Because of that, we also looked at the assigned clades in the GISAID database. According to data from the GISAID database, within a year of emergence, SARS-CoV-2 had evolved into nine clades, including L (to which virus reference strains belong), S, V, G, GH, GR, GV, GRY, and O [24]. The subsampled distribution of GISAID clades across Brazilian regions is shown in Figure 6. Overall, the clade GR (*n* = 16,142; 95.2%) was the most prevalent among the SARS-CoV-2 genomes submitted from Brazilian regions, followed by GRY (*n* = 375; 2.2%) and G (*n* = 245; 1.4%). Less common clades including V, GV, S, and L were identified in 0.04%, 0.02%, 0.02%, and 0.02% of the submitted genomes, respectively.

In addition, analysis based on the chronological distribution of SARS-CoV-2 clades in Brazil showed that clade G was predominant at the beginning of the pandemic. However, this could partially be an effect of the small number of sequenced genomes by then. Since this initial stage, the clade GR increased rapidly and stabilized as 74.5% of all sequences in March 2020, 89.2% in April 2020, and 86.9% in May 2020 and increased further to become the most frequent clade, with more than 90% in June 2020 (Figure 7).

### 3.3. Numbers of Genomes to Identify a New Variant

Regarding the published data obtained from GISAID, one question remains unclear. Is it possible to estimate how many genomes are necessary to identify a new mutant? The amount of sequences processed by countries is proportional to investment and laboratory resources. On the other hand, it is possible to mathematically equalize the obtained numbers of data and create a numeric value to correct it. We downloaded 1,629,158 sequences and grouped them as follows: South America, North America, Europe, Africa, Asia, and Oceania (Appendix A). Europe in total sequenced about 1,000,285 genomes of SARS-CoV-2, identifying 956 Pango lineages in 49 countries. The differences regarding genomic surveillance and viral spread are clearly inside Europe, where the United Kingdom sequenced 41.6% of all available genomes. Understanding the genome variability and evolution of genomes is fundamental to mounting an effective response to contain the pandemic; however, it requires governmental effort and scientific support. We already know that these difficulties are common in other countries, leading to a gap in being able to determine the hotspot locations by number of sequences. In Table 1 below, we calculated how many genomes are necessary to identify a new variant, considering the quantity of available genomes. The data (Appendix A show the estimated index and applied formulas) were equalized with a correction value estimated as 0.06485746617. The column G/L (genomes/lineages ratio) shows the data previously equalized. This analysis focused on the relevance of regions as hotspots for new variants, in cases of the number, G/L is smaller than the others, indicating a greater probability of finding a new mutant for each reported number. On the other side, the last column (Table 1) shows a diversity analysis between countries of each region, considering the maximum number of lineages of each country between geographical regions. This index, represented by alpha diversity (Table 1), shows the high diversity of lineages across Europe, numerically represented as 8.4996. In the range of each country, Europe varied from 0.0072 to the highest value 0.3676, represented by France. Second in diversity was Africa, reporting 6.4225, in which South Africa presented the highest rate (0.3381), followed by Asia at 5.9559. As for the Asiatic region, the model placed India (0.3677) as a hotspot for the generation of new variants, followed by Japan (0.3542). Across South America, the index was 3.4064, probably explained by the low availability of data and low proportion of cases in some countries. Analyzing each country in this region, we observed that according to genomes available Chile and Brazil represented the highest levels of variability (0.3679 and 0.3650, respectively). North America (1.6557) presented lower levels of diversity, and Canada was, surprisingly, a significant hotspot (0.3574). Oceania, as expected, represented a minimal level of estimated variation, showing an index of 0.9893.

## 4. Discussion

As SARS-CoV-2 continues to circulate in the human population after more than one year of pandemic, it is natural to observe genetic differences between SARS-CoV-2 strains sampled in various locations. This is the largest study focused on genome-wide mutational spectra covering nucleotides, amino acids, and deletion mutations in 16,953 complete and high coverage SARS-CoV-2 genomes from the five Brazilian geographical regions. Furthermore, this preliminary and crucial analysis of the Brazilian SARS-CoV-2 genomes shows an increase in the number of mutations.

Viral mutations are probabilistic events because of a viruses’ random transmission between infected people. Viral load is variable and depends on such factors as the course of infection and host immunity. Some individuals are “super spreaders”, which means that behavioral and environmental variables are relevant to infectivity, increasing successful transmission [25,26]. To date, we have around 20.8 million COVID-19 cases. Hence, here we are analyzing only ~0.05% of reported cases, comprising a snapshot of SARS-CoV-2 mutational status in Brazil.

A number of previous studies have examined variants within SARS-CoV-2 isolates; for example, [27,28] reported clustered groups of sequences showing geographical similarities, suggesting clusters of similar transmission in both time and viral strains. Resende et al. [29] showed that B.1.1.28 (E484K) is present in several states from the South, Northeast, and North Brazilian regions and dates its origin to 27 August 2020 (14 July–18 September). These findings documented a classical SARS-CoV-2 reinfection case with the emerging Brazilian lineage B.1.1.28 (E484K). Additionally, the authors provide evidence of this emerging Brazilian clade’s geographic dissemination outside the Rio de Janeiro state. Naveca et al. [30] reported a preliminary genomic analysis of SARS-CoV-2 B.1.1.28 lineage circulating in the Brazilian Amazon region and their evolutionary relationship with emerging and potential SARS-CoV-2 Brazilian variants harboring mutations in the RBD of spike protein. Phylogenetic analysis of 69 B.1.1.28 sequences isolated in the Amazonas state revealed the existence of two major clades that have evolved locally from April to November 2020 without unusual mutations in the spike protein. In Africa, Motayo et al. [31] showed the high prevalence of the D614 spike mutation (82%) between sequences analyzed.

More than identifying the mutations, these analyses allow continuous research focused on mapping how amino-acids changes affect antibody binding. In a recent study, Nonaka et al. (2021) [32] report the first case of reinfection from genetically distinct SARS-CoV-2 lineage presenting the E484K spike mutation in Brazil, a variant associated with escape from neutralizing antibodies. The mutations on the RBD domain enhanced ACE2 binding, promoting viral infectivity and maybe disrupting neutralizing antibodies (NAb) binding to evade the host immune response. Antibodies targeting RBD have been used and developed as therapeutics and are known as the major contributors to NAb responses. To control viral infection, a robust humoral immune response is essential in populations. Scanning mutations is important to map the RBD changes, and it is used to predict escape mutations in antibody epitopes [33,34].

Nevertheless, SARS-CoV-2 genomes sequenced in Brazil until now were clustered in at least nine major clades, as defined by the GISAID database. Of them, Clade GR was the most frequently identified in Brazilian genomes, followed by GRY and G. Since Clade G (Lineage B.1), defined by the spike protein’s D614G mutation, was identified, it rapidly predominated in many locales where it was found. Theoretical evidence suggests that mutations in the viral spike may be linked to altered potential for host cell membrane fusion, which should result in increased person-to-person transmission and pathogenicity [13,30,31]. A dub-cluster of clade G then started to split into GR, GH, and GV and more recently into GRY. Similar to what was observed recently, [24], within the analysis of the distribution of SARS-CoV-2 genomes across continents, showed that there was much expansion in the number of sequence genomes that were clustered into the GR and GRY clade compared with clade G, suggesting higher fitness for transmission by the newer clades compared with their ancestral one.

Here we evaluated the distribution of SARS-CoV-2 mutations across the five Brazilian geographical regions, showing different allelic frequencies with similar general distribution of all variants across different regions. We also showed the presence of 27 missense variants in the entire genome, the majority (44.4%) being present in the spike gene. Based on Pangolin software, we showed the presence of 61 SARS-CoV-2 lineages across Brazilian regions, with a high predominance of the Gamma variant. Based on Nextstrain clades, Brazilian genomes were classified into nine clades, with the majority belonging to clade 20B (*n* = 2724; 50.91%) and 20J (Gamma, V3) (2516; 47.21%). In GISAID clades, there are also nine clades, with a predominance of the GR clade (95.2%).

Finally, we estimated the number of genomes necessary to report a new variant and what is the ratio of this index by continent. The aim of diversity analysis was to show the hotspot regions in a printed scenario of pandemics. While we have a high genomic diversity in Europe given the large number of sequenced genomes, Africa is emerging as a hotspot for new variants. Asia is the third continent in terms of diversity. In South America, Brazil and Chile present mutation rates that are similar to South Africa and India. These numbers indicate that such regions are indeed hotspots for the emergence of new variants, especially when social restrictions are not applied strictly, leading to increased viral circulation. The genomic surveillance showed a potential tool for monitoring the circulation of SARS-CoV-2 and understanding the biological characteristics of the viral genome.

## Figures and Tables

**Figure 1 viruses-13-01806-f001:**
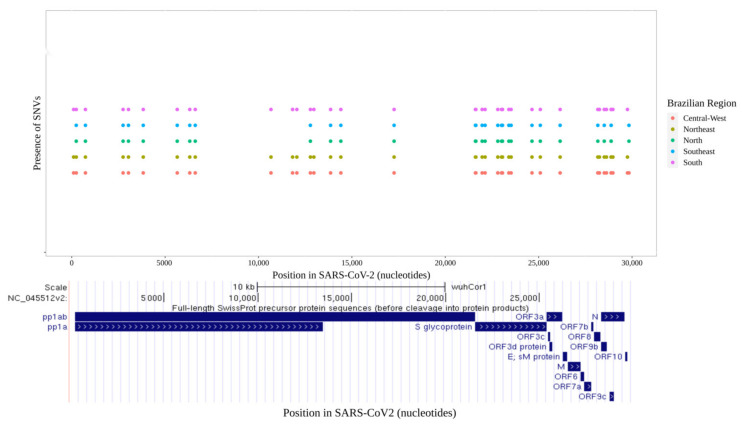
(**Upper**) Plot of all variants found by regions along SARS-CoV-2 nucleotide positions. (**Lower**) Snapshot of open reading frames (ORFs) of SARS-CoV-2.

**Figure 2 viruses-13-01806-f002:**
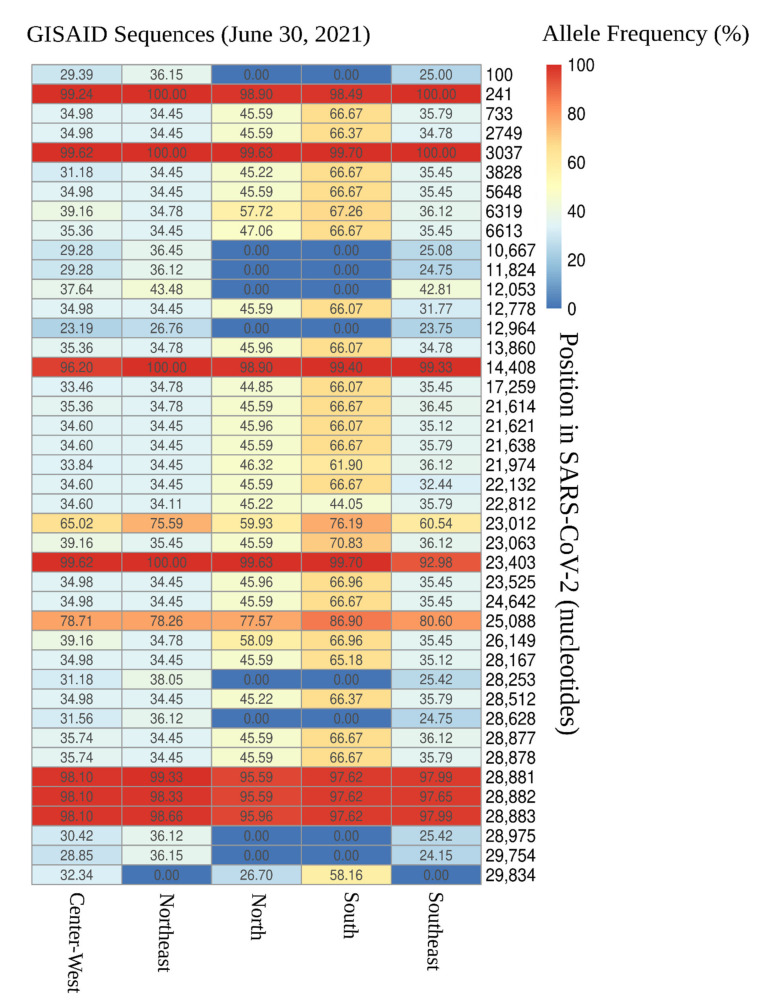
Allele frequency (plotted as percentage) of single nucleotide variants (SNVs) found in Brazilian SARS-CoV-2 high-coverage sequences on GISAID-EpiCoV.

**Figure 3 viruses-13-01806-f003:**
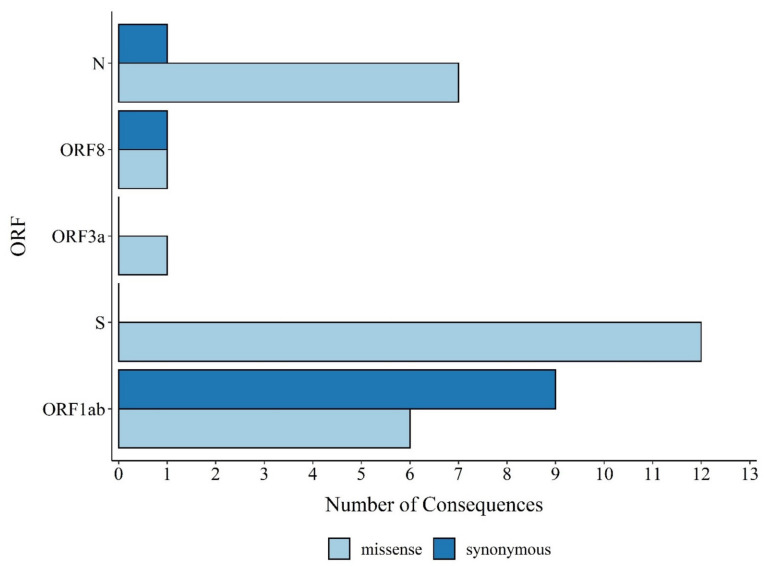
Missense and synonymous variant consequences per open reading frame (ORF) obtained with the Variant Effect Predictor.

**Figure 4 viruses-13-01806-f004:**
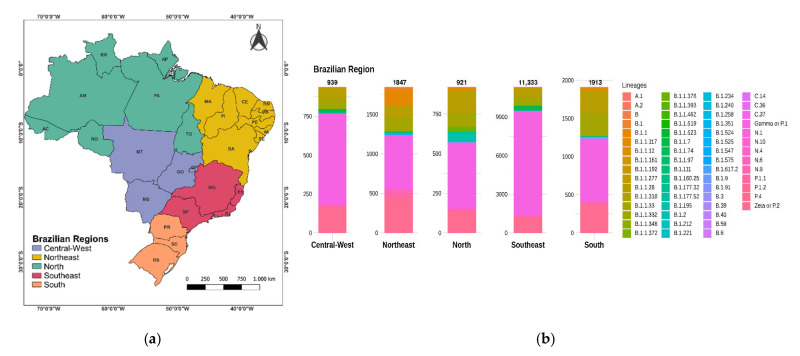
(**a**) Brazil’s regions. (**b**) Prevalent lineages across Brazilian regions, considering 16,953 genome sequences available in the GISAID database.

**Figure 5 viruses-13-01806-f005:**
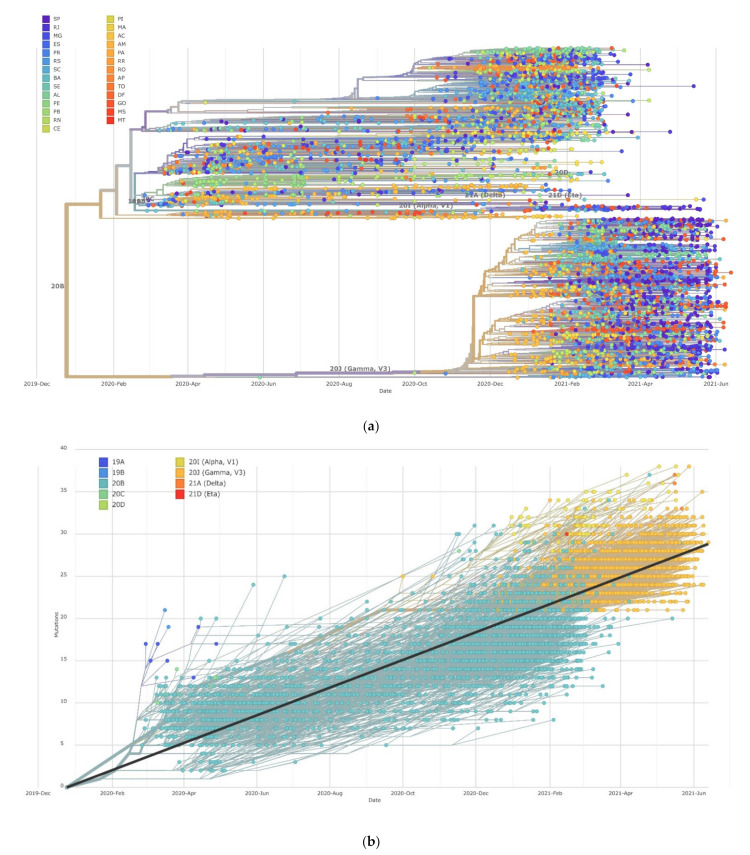
Phylogenetic tree and the clades assigned for the 5351 Brazilian SARS-CoV-2 genomes. (**a**) Colored by state. (**b**) Colored by clade.

**Figure 6 viruses-13-01806-f006:**
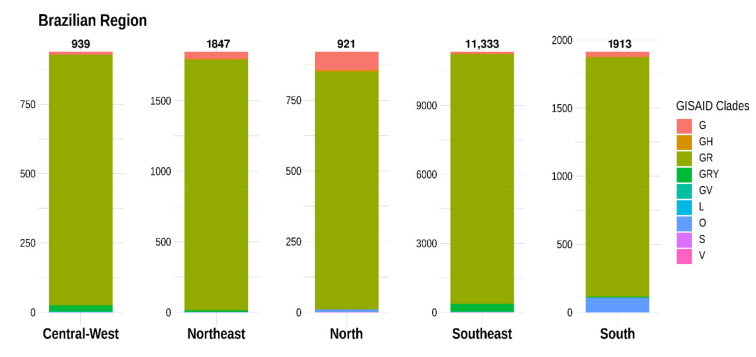
Subsampled distribution of GISAID clades across 16,953 Brazilian genome sequences.

**Figure 7 viruses-13-01806-f007:**
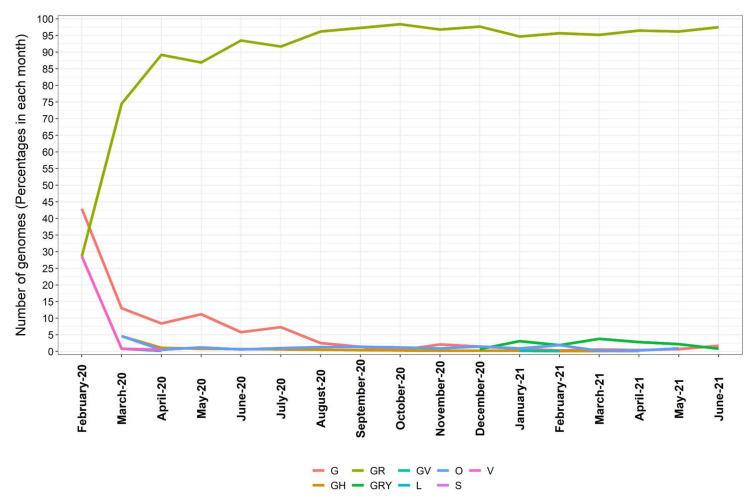
Chronological distribution of SARS-CoV-2 clades in the period from February 2020 till June 2021. Values expressed as percentages of the predominant clade in each month.

**Table 1 viruses-13-01806-t001:** Total number of genomes after filtering quality deposited and downloaded from GISAID. The data show predicted values for identifying one new variant as a function of sequences per region.

Region	*n* of Genomes	*n* Max of Lineages	G/L *	*n* of Countries	Diversity Region Index
South America	26,257	189	25.28	15	3.6906
Oceania	15,484	249	24.85	6	0.9893
Europe	1,000,285	956	293.21	49	8.4996
Asia	112,901	512	49.01	38	5.9559
North America	477,842	732	188.30	17	1.6557
Africa	11,873	239	13.99	38	6.4225

* G/L after applied correction factor.

## Data Availability

This study did not require the use of an ethics committee as we used genomes deposited in the GISAID database.

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
