# Peer review of "High Rate of Mutational Events in SARS-CoV-2 Genomes across Brazilian Geographical Regions, February 2020 to June 2021"

_viruses, 2021, doi:10.3390/v13091806_

Round 1

Reviewer 1 Report

In Viruses Manuscript ID# viruses-1344495, Souza et al. performed a meta-analysis of SARS-CoV-2 sequence diversity in Brazil from the beginning of the pandemic through June 2021.  They utilized publicly sourced databases to retrieve complete genome sequence data from all five Brazilian geographical regions and performed a comparative sequence analysis.  This manuscript provides interesting SARS-CoV-2 genomic surveillance information for Brazil that can be used to help understand the distribution and emergence of unique viral variants in the global pandemic.

The manuscript is well-written.  The figures provide an excellent display of the data, are clear and easy to interpret.  In addition to providing useful information about viral sequence divergence, the manuscript also presents interesting findings about the distribution of SARS-CoV-2 variants within the Brazilian population.  In particular, the striking emergence of the Gamma variant in Brazil is clearly presented.  This manuscript should be of significant interest to public health officials and epidemiologists monitoring the pandemic.  I have only 1 minor concern, listed below.  Once the authors have addressed that, I recommend acceptance of the manuscript for publication.

Minor Concern:

  1. I’m confused by lines 161-163 and Fig 2: Is the last region supposed to be South, not Southeast?  And why is this significant enough to highlight here in the results.  There are many variants displayed in Fig 2, what makes this SNV (or these regions) the most significant?

Reviewer 2 Report

This is a descriptive study of the SNVs characteristics for high quality SARS-CoV-2 genomes from Brasil sequenced so far, and the comparison of potential “real” genetic diversity of the virus in several world regions depending on the relationship between the number of sequenced viruses and the number of lineages/variants detected.

SPECIFIC COMMENTS:

  1. L72 and from here so on…the references in text do not match the reference list, check all the references.
  2. L97 include reference for Minimap2 aligner
  3. Figure 2 legend: I would substitute “founder variants in collected from…” for “ SNVs found in Brazilian SARS-CoV-2 high-coverage sequences on…”
  4. Figure 4. it will be clarifying for the reader to add a panel with a map depicting Brasil’s regions and states.
  5. Table S3 shold be simplified, perhaps by placing States in rows and lineages in columns, and then filling-up the corresponding squares when the lineage is present. Besides, you may want to include the region subdivision here as well.
  6. Figure 5: for panel A, maybe the coloring by region will be more easy to follow, so it matches figure 4 subdivisions.
  7. L360: “…genomes needed to detect a variant…”
  8. L365-367: Rephrase to “In hotspot regions to emerging variants emphasis on vaccination is needed.”
  9. L367-371: I recommend deleting these two last sentences.

MINOR POINTS:

  1. Delete “Africa is emerging as a hotspot for new variants”
  2. Transmembrane domain
  3. L105: “…nextstrain platform…”
  4. L112: Reference for Augur: 21?
  5. L116: “…submitted to the GISAID database….”
  6. L124: “…(i) to calculate an estimate….”
  7. L276: specify G/L=genomes/lineages ratio
  8. L284: “…placed India…”
  9. L259: delete “for the authorss”
  10. L326: “…mutation (62%) between…”
  11. L152 “…sequenced regions…”
  12. L187: “…tehe envelope…”
  13. Table S1: “ΣG/n maximum?? Lineages”
